# Convergence and divergence in mortality: A global study from 1990 to 2030

David Atance[1]ⓘ*, M. Mercè Claramunt[2]ⓘ, Xavier Varea[2]‡, Jose Manuel Aburto[3,4]‡

**1** Departamento de Economía y Dirección de Empresas, Universidad de Alcalá, Madrid, Spain,
**2** Departamento de Matemática Económica, Financiera y Actuarial, Universitat de Barcelona, Barcelona,
Spain, **3** London School of Hygiene and Tropical Medicine, London, United Kingdom, **4** University of Oxford,
Oxford, United Kingdom

☉ These authors contributed equally to this work.
‡ XV and JMA also contributed equally to this work.
* david.atance@uah.es

journal.pone.0295842

LTD, INDIA

**Data Availability Statement:** The data underlying
the results presented in the study are available
from https://population.un.org/wpp/.

**Funding:** This work was partially supported by the
funded chair UB-Longevity Institute. The funders

## Abstract

An empirical question that has motivated demographers is whether there is convergence
or divergence in mortality/longevity around the world. The epidemiological transition is the
starting point for studying a global process of mortality convergence. This manuscript aims
to provide an update on the concept of mortality convergence/divergence. We perform a
comprehensive examination of nine different mortality indicators from a global perspective
using clustering methods in the period 1990-2030. In addition, we include analyses of pro-
jections to provide insights into prospective trajectories of convergence clubs, a dimension
unexplored in previous work. The results indicate that mortality convergence clubs of 194
countries by sex resemble the configuration of continents. These five clubs show a com-
mon steady upward trend in longevity indicators, accompanied by a progressive reduction
in disparities between sexes and between groups of countries. Furthermore, this paper
shows insights into the historical evolution of the convergence clubs in the period 1990-
2020 and expands their scope to include projections of their expected future evolution in
2030.

## 1 Introduction

Most countries in the world have seen sizable improvements in longevity over the last two cen-
turies [1]. While countries differ in their trajectories and levels of mortality improvements,
many similarities, such as the shift from high infant mortality to profiles where older ages are
more important, have been found [2–5]. Hence, an empirical question that has motivated sub-
stantive research is whether there is convergence or divergence towards a unique pattern of
mortality and longevity across countries [3–10]. Some authors have tried to establish a univer-
sal pattern of mortality change, e.g. the epidemiological transition theory [11], but some atypi-
cal trajectories of declines in life expectancy observed in countries such as post-Soviet nations
or the HIV crisis in sub-Saharan Africa have challenged these universal theories, as well as
cause-of-death profiles at shared levels of life expectancy [12]. Although we cannot truly speak
of global convergence, groups of countries have followed very similar processes of mortality

had no role in study design, data collection and analysis, decision to publish, or preparation of the manuscript.

**Competing interests:** The authors have declared that no competing interests exist.

change [2, 5, 13, 14]. For example, longevity has been classified across countries based on proximity and geographical locations, as well as socioeconomic and environmental conditions [4, 6, 8, 13, 15]. As [3] puts it, "There is not a single mortality system and is cut by deep faults into a number of blocs, each with its own distinctive trajectory."

Previous studies on the analysis of convergence/divergence focused on life expectancy at birth as an indicator of longevity. Life expectancy at birth indicates the average years a cohort of newborns is expected to live if they experience the observed death rates in a year throughout their lifespans [16]. Life expectancy is a valuable measure of population health that reflects the cumulative social, economic, medical, and technological achievements of society [17–20]. It is used in international comparisons because it is not affected by population size or population age structure [21–25], and it is correlated with economic growth [17, 26–28]. However, while life expectancy is an important indicator, trends in life expectancy mask changes and variations underlying schedules of mortality [15, 24, 29]. Complementary indicators of life expectancy can provide insights into other aspects of mortality change. For example, [30] classify several mortality indicators under the following categories: central longevity, horizontalization, concentration and/or verticalization, rectangularization, and maximum longevity.

The objective of this paper is to update the analysis of convergence/divergence of patterns of mortality. We have performed a cluster analysis of 194 countries by sex. Countries represent all continents and mortality profiles are very diverse. Additionally, we have complemented previous analyses by incorporating indicators of mortality change other than life expectancy, following previous ideas about clustering countries into convergence clubs [3–5, 28, 31]. We have applied a state-of-the-art methodology to group countries based on multiple characteristics of mortality over time using an in-sample and out-of-sample approach. We have shown how mortality convergence/divergence evolves in three specific periods: two using the in-sample approach in 1990 and 2010, and one using the out-of-sample approach for 2030. The idea is to study how mortality convergence/divergence is characterized and how is expected to evolve in the future.

Specifically, we have complemented life expectancy with modal age at death, Gini index, standard deviation conditional to surviving to different ages, and the percentile of the number of people lived. We have used modal age at death to have a different perspective of the changes in the distribution of deaths and to explain the change in mortality at older ages [30, 32, 33]. Modal age at death has become an increasingly important indicator in low-mortality countries [33]. In addition, to capture variation in the length of life, we included the Gini coefficient and the standard deviation [34, 35]. Lifespan variation indicates the uncertainty in the timing of death at the individual level [29, 36], and it is an indicator of heterogeneity in underlying population health at the population level [29, 36]. Finally, we have complemented this analysis with two percentiles, which represent the survival age of the percentage of individuals who reach a given age. This percentile in the actuarial field is also known as "life preparancy" [37]. This indicator can measure the degree of horizontalization of the survival curve.

The remaining sections of the paper are organized as follows. In Section 2, we present the database that we used and provide a brief overview of the [38] mortality model to forecast age-specific probabilities of death. In Section 3, we define the mortality indicators that are used to cluster mortality around the world and through the years. In Section 4, we introduce the clustering method to identify consistent country groups. In Section 5, we show the clusters identified around the world. Finally, in Section 6, we discuss and conclude the results of the paper.

## 2 Data and methodology

### 2.1 Data

This study uses data by sex for the period 1990–2020 from [39]. This database provides accessible demographic data for all countries and areas of the world. Except Channel Islands, Eswatini, French Guiana, Guadeloupe, Martinique, Mayotte and Réunion (countries with a reduced population number, and most being islands). Various other sources could be employed such as [40], we carry out an ambitious study around the world using the remaining 194 countries which contains mortality data from European countries, USA, Canada, Chile, Israel, Japan, New Zealand, South Korea and Taiwan, and Latin American Human Mortality Database [41] including information for Latin American countries. These two databases contain high-quality mortality information from the census and the death registry of the national statistical institute of each respective country. However, to maintain consistency across all the countries included in the study and to achieve large geographic coverage, even at the cost of using data of questionable quality, we decided to use a single database. Table 1 in S1 Annex includes the list of the analyzed countries and their acronyms. It should be noted that the analysis of convergence and divergence in mortality was initially done with three different databases. The clustering method is inclined to group countries in the same club/cluster from the same database. Therefore, to achieve greater consistency, we decided to use data from a single source.

It is essential to recognize that the United Nations Populations Division (WPP) seeks to generate reliable estimates of mortality and lifetables. Specifically, the WPP incorporates the original mortality rates published by each country's national statistical institute from births, vital registrations, and deaths. Nonetheless, in populations where such data from vital registration are not available or of insufficient quality, WPP uses model life tables for the estimation of mortality rates. In 2022, WPP employed five model life tables known as CD North, CD South, CD West, HIV/AIDS, and LogQuad models. For a deeper understanding of the construction of these five life tables, the reader may refer to [42, 43]. Furthermore, it is worth emphasizing that the WPP methodology has evolved over time, incorporating new techniques. For instance, in 2022, Bayesian hierarchical [44] were integrated to estimate mortality rates between the ages of 15 and 60.

All mortality data from [39] are grouped by age (children under 1 year, between 1 year and 5 years, and then by groups of 5 years, the last group being 100 years or more) and by periods of 5 years, i.e. abridged life tables. For model adjustment, we used information from the five-year period 1990–1995 to the last five-year period available, 2015–2020, and up to the group of +99 years old. We have included all the groups from the WPP life tables. However, there are few countries with no information in the last group; in these populations, we have used the information of the previous group. We have chosen to assign the data for the five-year period to the year at the lower end; thus, the data assigned to 1990 corresponds to the five-year period 1990–1995. With these data, life tables were constructed following standard procedures [45]. It should be noted that some countries may have poor quality data, especially those countries with low incomes and/or with conflicts as wars or deleterious socioeconomic and political conditions.

### 2.2 The Lee-Carter model

There is a broad variety of models that have been introduced in the actuarial and demographic literature to model and forecast the probabilities of death. It is challenging to predict and model age-specific probabilities of death because there is no widely accepted best method [46].

One of the most used models is the original work of [38]. Since its publication in 1992, it has been widely used for mortality trend fitting and forecasting. This is due to the simplicity of the estimation, the easy interpretation of its parameters and its parsimony [47, 48]. For these reasons, we have decided to use the [38] model to fit and forecast the probability of death. Furthermore, [46] show that [38] model has better forecasting ability than a battery of other mortality models in 30 European countries employing different re-sampling methods.

We used the logit link version of the model for death probabilities, $q_{x,t}$ [49]. By applying this methodology, firstly we are guaranteed to obtain $q_{x,t}$ not greater than one [47] and, secondly, we maintain the historical links with the early actuarial works of [50]. The estimation of the model parameters is carried out using the **gnm** library [51] from [52], which allows to obtain the parameters by maximum likelihood.

Once the model has been fitted, the mortality forecast of the age-specific probabilities of death, $q_{x,t}$, are obtained by modeling $k_t$, an index that describes the general tendency of mortality over time, as a time series. We assume that $k_t$ follows an Autoregressive Integrated Moving Average model (ARIMA) independent process. To estimate the ARIMA parameters, we used the **auto.arima** function from the *R* package *forecast* [53], which provides the ARIMA model that offers the best results, according to the Akaike information criterion (AIC).

## 3 Mortality indicators

For each year studied using the in-sample and out-of-sample approach, we have computed different mortality indicators: life expectancy at age 0 and 65, modal age at death, Gini index at age 0 and 65, conditional standard deviation at age 0 and 65 and two percentiles. Following [30], life expectancy and modal age at death are classified as central longevity indicators; conditional standard deviation and Gini index are indicators of concentration; percentile is a mapping indicator. The steps that are necessary to estimate all mortality indicators, including life expectancy, are shown in S2 Annex.

### 3.1 Life expectancy

Life expectancy at $x$ and time $t$, $e_{x,t}$ represents the average age at death if an individual experienced the prevailing death rates at time $t$ throughout their lifetime from age $x$ [54, 55]. It can be calculated for any age. We used $e_{0,t}$ and $e_{65,t}$ to denote life expectancy at birth and at age 65, respectively. By including both indicators in our analysis, we aim to capture the different dimensions of the mortality phenomenon in the different age groups. While $e_{0,t}$ provides the impact of mortality in the infant and child ages, $e_{65,t}$ examines the mortality patterns among the elderly and minimizes the extrinsic factor associated with infant mortality [56]. Together, these two mortality indicators contribute to a more comprehensive assessment of mortality convergence and divergence across the world.

### 3.2 Modal age at death

Modal age at death is denoted by $M_t$, and it is the age at which the highest number of deaths occur. In this paper, we use abridged life tables, so the modal age of death corresponds to the mean age of the interval with the highest number of deaths. This indicator is used to study the changing mortality scenario experienced by countries with low mortality. Furthermore, the study of modal age at death gives the opportunity to have a different perspective on the changes in the distribution of deaths and to explain the shift in mortality at older ages [33]. $M_t$ is selected because it may reveal changes in the probability of death $q_{x,t}$ that are not measured with life expectancy [54].

### 3.3 Gini index

The Gini index is a measure of the inequality of lifespans and summarizes the Lorenz mortality curve in a single value [34]. It takes values between 0 and 1; the value zero corresponds to an equal distribution of years of life, which means that all individuals die at the same age. Meanwhile, the value 1 corresponds to the extreme, when all individuals die at the earliest age, except for one, who dies at the highest attainable age. [57] highlight that it is a robust standardized dimensionless index, as it is not affected by a uniform decrease/increase in mortality across age groups. In this study, the Gini mortality coefficient at birth is denoted by $G_{0,t}$ and the one corresponding to age 65, by $G_{65,t}$.

### 3.4 Conditional standard deviation

Life expectancy collects all the information about the expected life years of an individual, but two populations with the same life expectancy can have very different mortality distributions. The conditional standard deviation is an absolute measure of dispersion that allows to show whether life expectancy is representative (all individuals survive exactly as life expectancy indicates) or poorly representative (the differences regarding life expectancy are very high, not all individuals die at the same age) of the population in which it is calculated [8]. We have calculated at age 0 and 65, which are denoted by $s_{0,t}$ and $s_{65,t}$ respectively.

### 3.5 Percentile

Let $y_{p;x,t}$ be the pth percentile of $l_{x,t}$, the number of individuals that are still alive at age $y$ at time $t$ given that they were alive at age $x$. Thus, $y_{p;x,t}$ is equal to an age. In fact, this percentile is a measure of the degree of "horizontalization" of the survival curve [30], and is formally defined in terms of the cumulative density function and its complement, the survival function. This indicator helps to prepare the retirement portfolios to succeed 90 or 95 percent of the time, rather than 50 percent of the time, as implied life expectancy. In this paper, we calculate two percentiles: $y_{50\%;0,t}$, and $y_{75\%;65,t}$.

### 3.6 Correlation between the mortality indicators

In this section, we analyze the correlation between the 9 mortality indicators ($e_{0,t}$, $e_{65,t}$, $M_t$, $G_{0,t}$, $G_{65,t}$, $s_{0,t}$, $s_{65,t}$, $y_{50\%;0, t}$, and $y_{75\%;65,t}$) used in this manuscript. Fig 1 presents the results. There are positive and highly (higher than 0.85) correlations between $e_{0,t}$, and $e_{65,t}$, $y_{50\%;0,t}$, and $y_{75\%;65,t}$. This outcome can be attributed to the fact that $e_{0,t}$ requires all the mortality rates for its estimation, while the other indicators use exclusively the end of the mortality curve. Consequently, countries with lower infant and young mortality rates tend to exhibit a high correlation between these indicators because the changes at the end of the mortality affect both indicators. Furthermore, $e_{0,t}$ is inversely and strongly (lower than −0.85) correlated with $G_{0,t}$ and $s_{0,t}$. This association arises because lower values of $G_{0,t}$ and $s_{0,t}$ indicate better results, while $e_{0,t}$ is the opposite, i.e., higher values indicate better results. The remaining correlations between the indicators can be observed in Fig 1.

We emphasize that all mortality indicators have been included in our study in order to cover different perspectives of the mortality phenomenon, despite the high positive and negative correlation between the mortality indicators. Indeed, when we performed a parallel analysis employing only $e_{0,t}$, $M_t$, $G_{0,t}$, and $G_{65,t}$, the clusters and number of countries in each cluster differed from the results presented in this manuscript. This fact underscores the value of incorporating a range of mortality indicators, as their inclusion leads to capturing the mortality phenomenon from all perspectives.

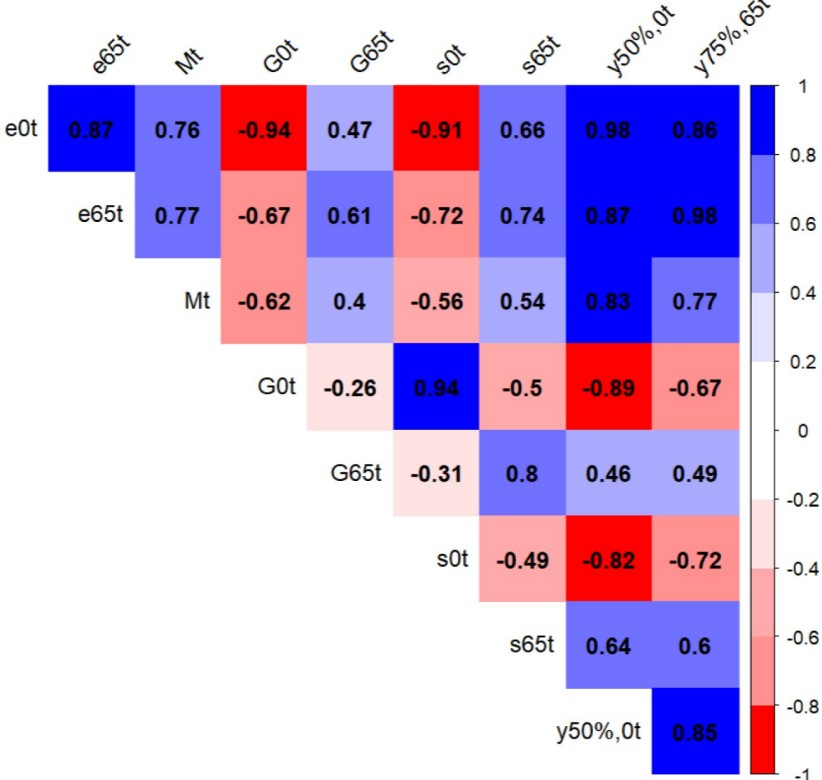

**Fig 1. Correlation plot between all mortality indicators used in the study, including both sexes and periods.** The image was created using *corrplot* R-package, developed by [58].

## 4 Cluster methodology

We use a clustering method that has the purpose of finding patterns or subgroups (clusters) among a set of observations [59]. This technique is considered a multivariate statistical analysis since it is a statistical analysis of data containing observations in which more than one variable is measured [60]. The cluster methodology allows us to split the mortality indicators into different groups, so countries within each cluster are quite similar to each other, while countries in different clusters are quite different from each other. In this paper, we have grouped the countries according to the nine indicators explained in the previous section.

Within the multitude of existing clustering methods, we used the hierarchical K-means for its ease of interpretation and good results with a large number of data and variables [59, 61]. The Euclidean distance is used to measure the similarities or differences between the countries, while the linkage average is employed to calculate the distance between clusters. It is calculated as the mean difference between all indicators within two clusters. The Euclidean distance between two countries is the length of the line segment connecting them [62] that is the gap between the countries. We used it because it is the most common distance employed in published mortality papers about clustering methods [54, 62–64]. The clustering process was carried out using the following libraries: **factoextra** [65], **cluster** [66] and **mclust** [67], which allowed us to obtain all the results in this paper.

## 5 Results

### 5.1 Clustering evolution

In this subsection, we present the cluster results for three specific years for females and males: 1990 (Table 4 in S3 Annex), 2010 (Table 5 in S3 Annex), and 2030 (Table 6 in S3 Annex). We restricted the results to these three periods to show how groups are formed and whether there is convergence/divergence around the world. We also grouped the countries in the other periods (1995, 2000, 2005, 2015, 2020, 2025) and the data results are available from a public repository at https://github.com/davidAtance/Longevity-Paper. We used five clusters as the optimal number of groups in all periods and sexes. We use several clustering indices, including the Silhouette, Hartigan, . . . indices. In fact, we use the **NbClust** R-package to select the optimal number of clusters and five was the optimal number. In the next subsections, we show that the countries belong to different clusters in the three years of analysis. Tables 2 and 3 in S3 Annex summarize this information. In order to characterize the clusters, each table includes a central value (C) corresponding to the centroid of the cluster, which represents the mean value of countries in each cluster [68], a minimum (m) value, and a maximum (M) value of the nine indicators described in section 3. We also indicated a representative country (R.C.), where indicators have the lowest Euclidean distance to the cluster centroid. In fact, we present in Table 7 of S3 Annex the mortality indicators or each centroid of each cluster and period.

**5.1.1 Male population.**   Fig 2 shows results for 1990. The male Cluster 1 (MC1–1990) includes 31 countries, mostly from Sub-Saharan Africa except the southern cone, and its R.C. is Guinea-Bissau. The male Cluster 2 (MC2–1990) includes 54 countries, most of them OECD countries, and its R.C. is the United States. MC3–1990 is the largest (64 countries) and consists of South America, North Africa, former Soviet republics, the Middle East, and China, and its

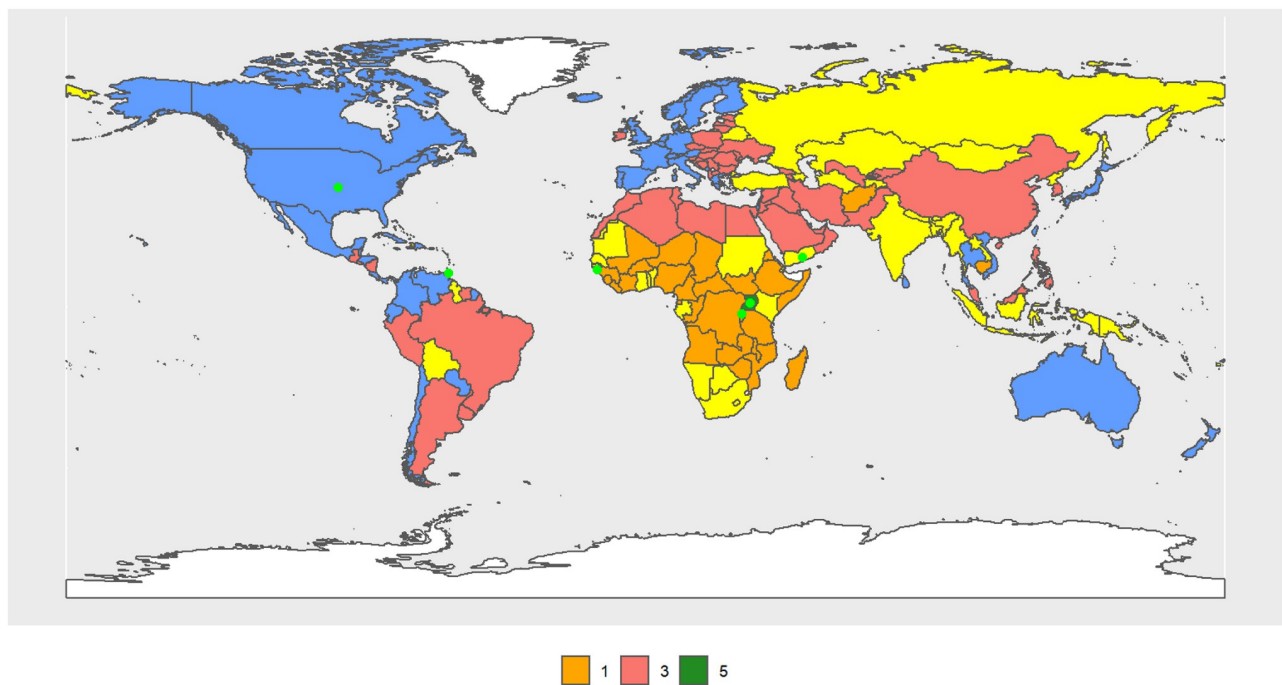

**Fig 2. World map of 1990 male clusters.** The map was created using *rnaturalearthdata* R-package, developed by [69].

R.C. is Trinidad and Tobago. Russia, India, Mongolia, the countries of southern Africa, and some Pacific islands, up to a total of 43 countries, form the MC4–1990 and its R.C. is Yemen. Rwanda and Uganda form the MC5–1990.

Considering only the central values of the indicators in each cluster, MC2–1990 and MC3–1990 always occupy the first and second positions in life expectancy, mode, and age of preparation for life. Then, in general, MC2–1990 takes the largest values for the above indicators, and MC5–1990, the smallest values. In contrast, the order is reversed for inequality and standard deviation at birth.

In 2010, the configuration of the clusters tend to resemble the continents. MC1–2010 includes practically all of Sub-Saharan Africa (52 countries) and its R.C. is Uganda. MC2–2010 has lost some countries from the Caribbean and Central America but essentially continues to be formed by the OECD (37 countries). Its R.C. is Austria. MC3–2010 is basically formed by South and Central American countries plus Poland, Algeria, and some Pacific islands. Its R.C. is Thailand. MC4–2010 is formed by countries that came from MC3–1990 and MC4–1990. Its R.C. is Grenada. MC5–2010, just as in 1990, was formed by only one country, Lesotho. Fig 3 shows the volume of the transitions, while Table 2 in S3 Annex and Figs 2, 4 and 5 show the cluster changes in the countries.

Considering only the central values of the indicators in each cluster, MC5–2010 and MC1–2010 always occupy the last and penultimate positions, respectively. Clusters MC2–2010, MC3–2010, and MC4–2010, share the first three positions, with the most usual order being MC2–2010 > MC3–2010 > MC4–2010. As was the case in 1990, the order is reversed for inequality and standard deviation at birth.

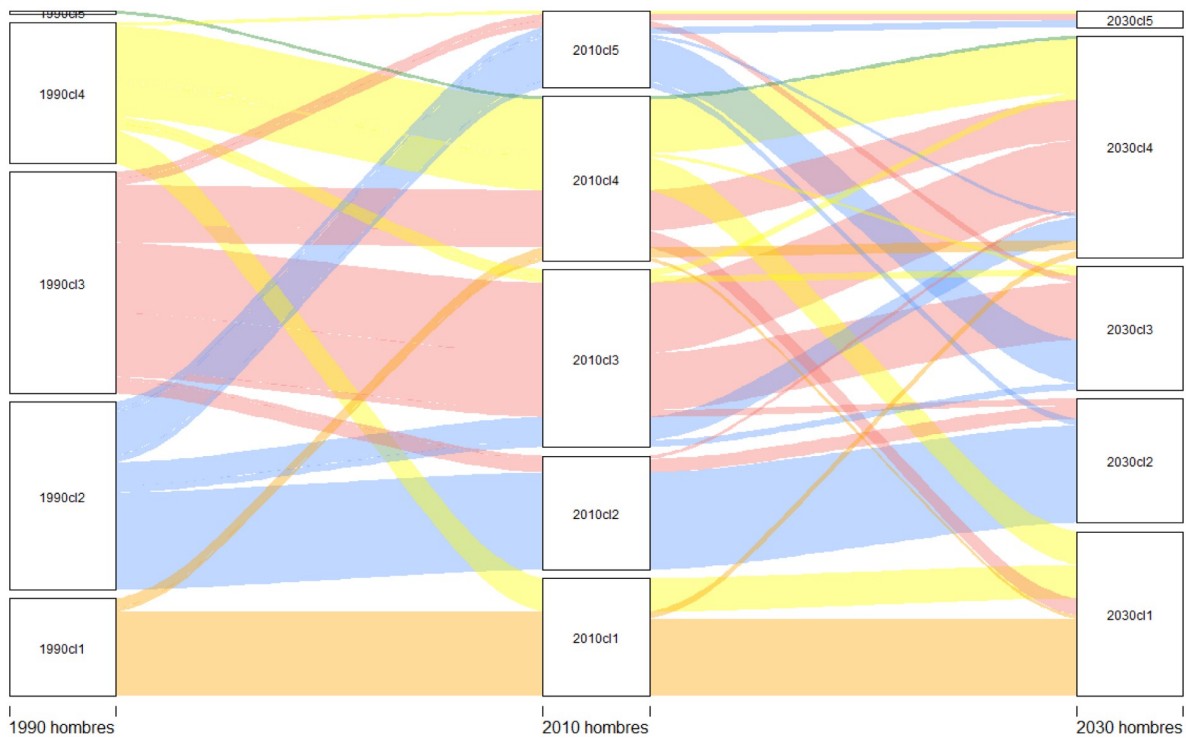

**Fig 3. Diagram to show how the convergence groups (clusters) change their male populations around the world over the period of 1990–2010 and 2010–2030.** The image was created using *alluvial* R-package, developed by [70].

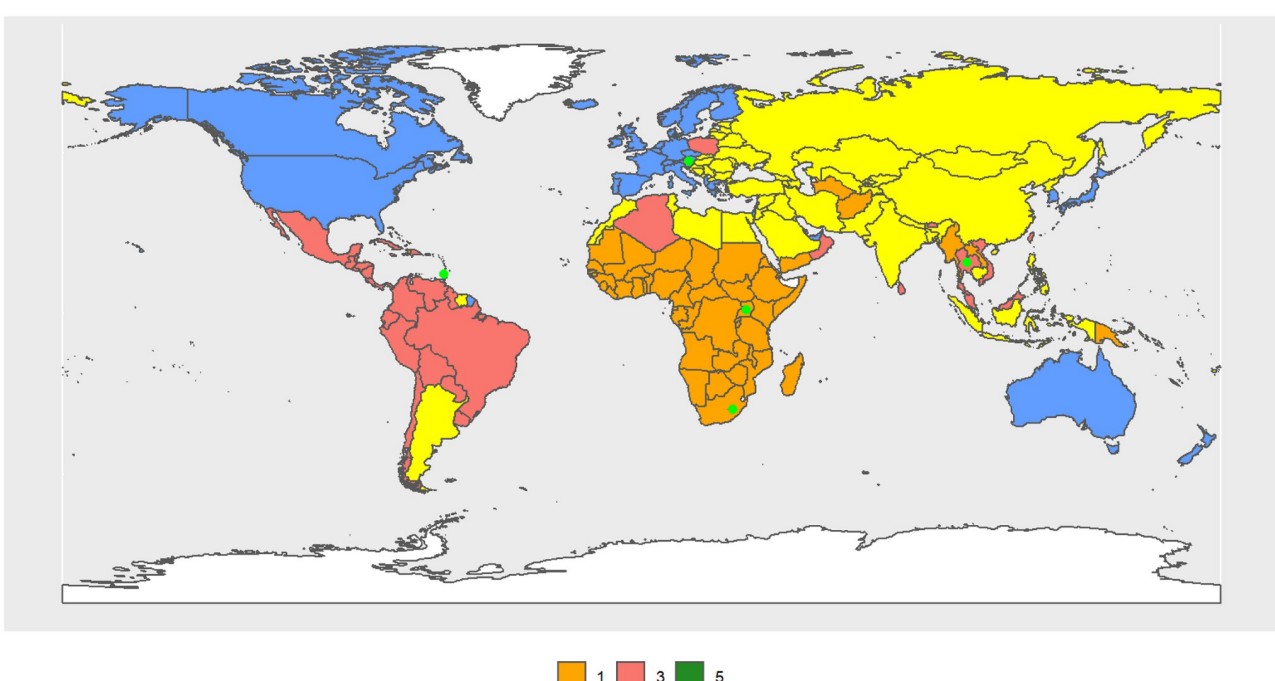

**Fig 4. World map of 2010 male clusters.** The map was created using *rnaturalearthdata* R-package, developed by [69].

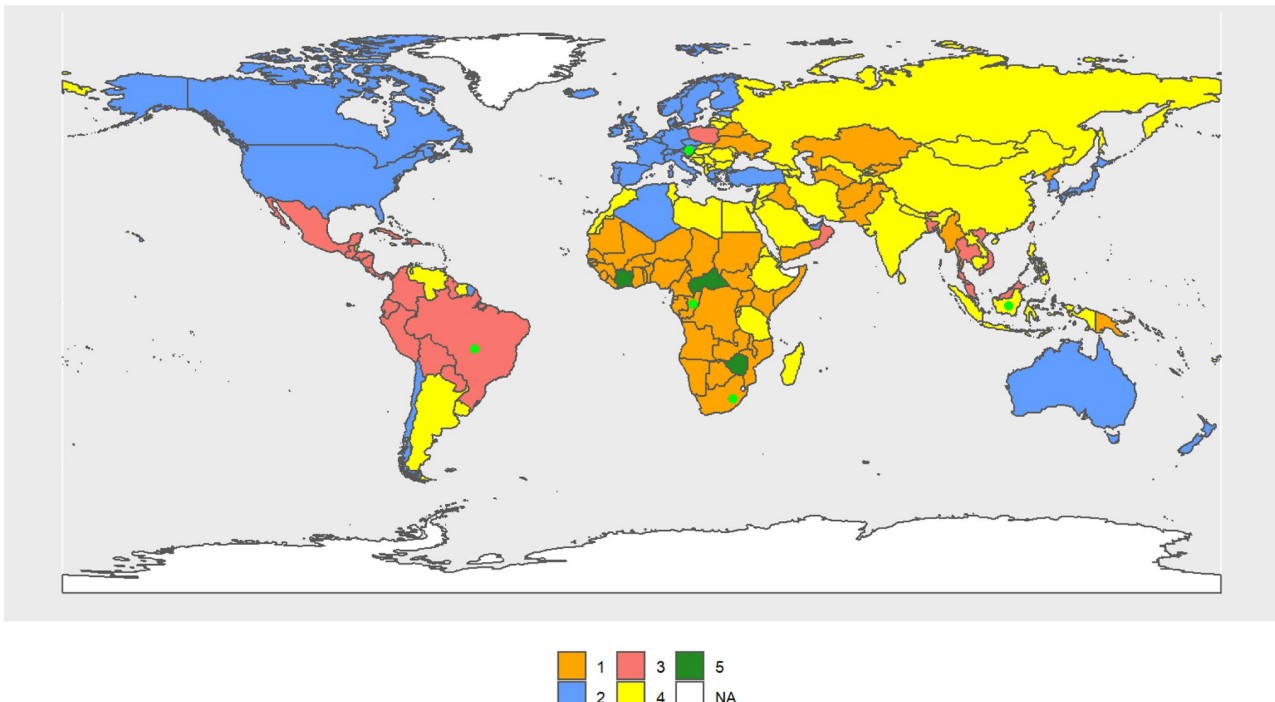

**Fig 5. World map of 2030 male clusters.** The map was created using *rnaturalearthdata* R-package, developed by [69].

In 2030, the cluster configuration continues to resemble the continents (see Fig 5). MC1–2030 includes Africa and its R.C. is the Republic of Congo; MC2–2030 is formed by North America, Europe (except the Eastern countries) and Japan, Australia, New Zealand, and Chile, and its R.C. is Austria; MC3–2030 includes Central and South America, Poland, and several Pacific Islands, and its R.C. is Brazil; MC4–2030 is formed by Asia and North Africa, and its R.C. is Indonesia; and MC5–2030 includes four countries from Africa (the Central African Republic, Cote d'Ivoire, Lesotho and Zimbabwe), and its R.C. is Lesotho. It is precisely these four countries that have the lowest values for $e_{0,t}$, $e_{65,t}$, $M_t$, and $y_{50\%;0,t}$.

In addition, we have introduced Fig 6, which presents the geographical proximity of each country for males within each period based on the two principal components obtained through the Principal Component Analysis (PCA). This visualization allows us to depict the distance between the countries within each cluster for every period and, also, allows us to explain why countries move from one cluster to another in the next period, as a result of the distance to the next cluster. Notably, we observe a distinct pattern in MC5–1990, MC5–2010, and MC5–2030 where an independent cluster is grouped using two, one, and, four populations, respectively, with considerable distance from the remaining clusters.

**5.1.2 Female population.** The female Cluster 1 (FC1–1990) (see Fig 7) includes 29 countries, mostly from Africa, as was the case for males, and its R.C. is the Central African Republic. The female Cluster 2 (FC2–1990) includes 53 countries, mostly from the OECD, and its R.C. is Barbados. FC3–1990 (68 countries) consists of the largest cluster, mainly made by North Africa, South America, the Pacific Islands, Russia, and former Soviet republics, and its R.C. is Saint Vincent and the Grenadines. India, South Africa, and the Pacific Island, up to a total of 42 countries, form FC4–1990 and its R.C. is Tajikistan. Rwanda and Qatar form FC5–1990.

Considering only the central values of the indicators of each cluster, FC2–1990 > FC3–1990 > FC4–1990 > FC1–1990 > FC5–1990 in almost all indicators. Then, FC2–1990 always takes the largest values for the above indicators, and FC5–1990, the smallest values. In contrast, the order is reversed for inequality and standard deviation at birth.

In 2010, clusters are configured as follows: FC1–2010 (54 countries) now also includes Rwanda and South African countries, which were part of FC5–1990 and FC4–1990, respectively. Its R.C. is Uganda. FC2–2010 has lost the Central and South American countries and is essentially made up of the OECD (41 countries). Its R.C. is the Netherlands. FC3–2010 is basically made up of Central and South American countries (25). Its R.C. is Vietnam. FC4–2010 is made up of Asian countries (previously in FC3–1990), North Africa, and the Middle East (73 countries). Its R.C. is Jamaica. FC5–2010 consists of one country. Its R.C. is Lesotho. Fig 8 shows the volume of transitions, while Table 3 in S3 Annex and Figs 7, 9 and 10 show the cluster changes in the countries.

Considering only the central values of the indicators in each cluster, FC2–2010 > FC3–2010 > FC4–2010 > FC1–2010 > FC5–2010 in life expectancy, mode and percntile.

In 2030 (Fig 10), FC1–2030 and FC2–2030 remain essentially the same, with Sub-Saharan Africa and OECD countries, respectively. FC1–2030 has 42 countries and its R.C. is South Africa, while FC2–2030 has 42 countries and its R.C. is Ireland. The composition of the remaining clusters is highly different from those of 2010. FC3–2030 (11 countries) has R.C. Dominican Republic, FC4–2030 (61 countries) has as R.C. São Tomé and Principe. FC5–2030 consists of 38 countries. Its R.C. El Salvador.

The characteristics of the clusters indicate that longevity in the male population is more homogeneous among the different countries than in the female population. The coefficient of variation between the different countries for each of the indicators and their evolution over time confirms this. The coefficient of variation in the three years analyzed and for all indicators is lower in the male population (except GI65 in 2030).

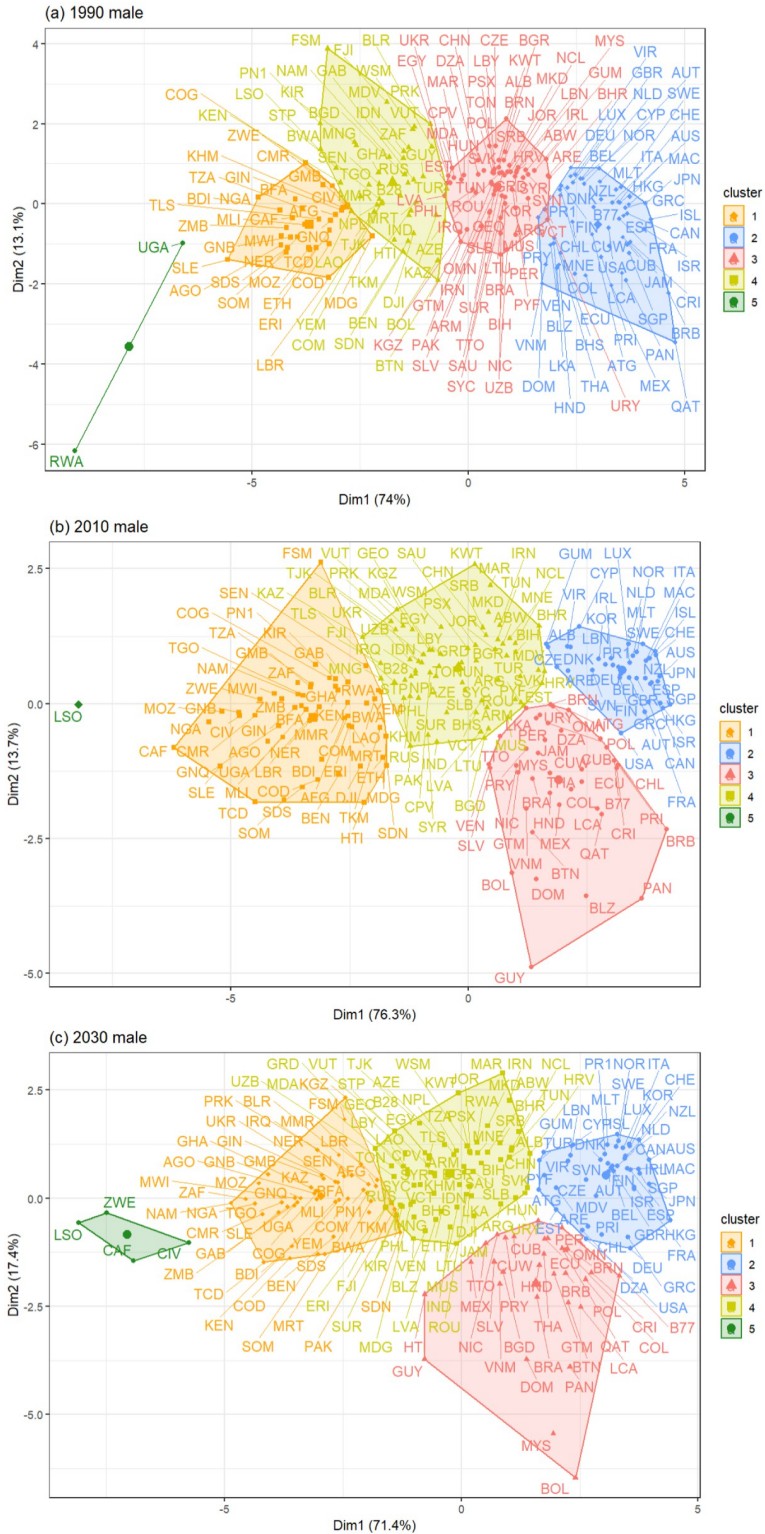

**Fig 6. Geographical proximity between the two principal components in (a) 1990, (b) 2010, and (c) 2030 for the mortality indicators in males using PCA.** The image was created using *factoextra* R-package, developed by [65].

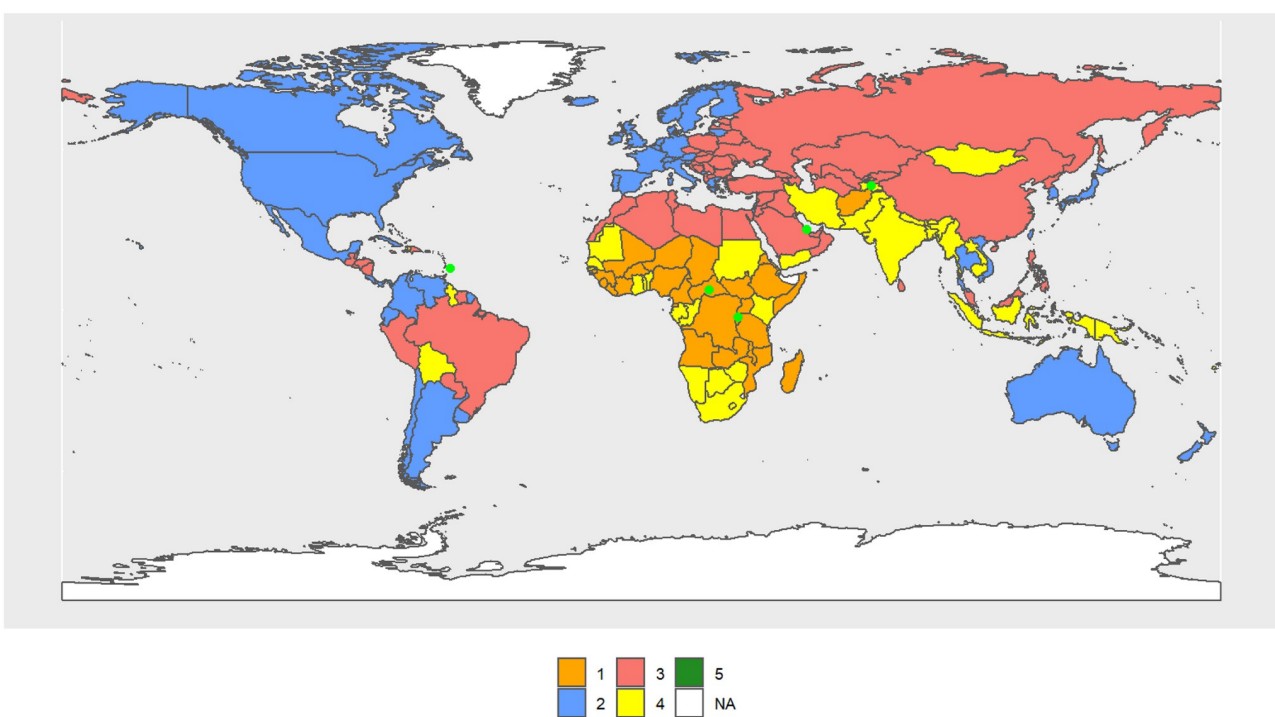

**Fig 7. World map of 1990 female clusters.** The map was created using *rnaturalearthdata* R-package, developed by [69].

Additionally, we have introduced Fig 11, which presents the geographical proximity of each country for females within each period based on the two principal components obtained through the Principal Component Analysis (PCA). Notably, we observe a distinct pattern in FC5–1990 and FC5–2010, where an independent cluster is grouped using two and one population, respectively, with a considerable distance from the remaining clusters.

## 6 Discussion

In this study, we presented a global analysis of convergence/divergence in mortality from two different points of view: a temporal evolution, and a geo-economic perspective by sex. Indeed, new procedures, such as clustering methods, which are encompassed within machine learning techniques, have been used to try to capture convergence clubs of mortality around the world. It should be mentioned that this tool has been employed to group mortality in a reduced number of countries and using a lower number of mortality indicators compared with this study; see for instance [54, 62–64, 71, 72].

[5], when considering the health transition, chose to divide the countries of the world into five convergence clubs for the period 1950–2010. In this paper, using a sophisticated statistical procedure (clustering methods), we used the same number of convergence groups for 1990–2030. These five clubs are consistent with those selected by [5]. In 1990, the five groups, regardless of sex, were as follows: 1. Central African countries, which are considered low-income (MC1–1990 and FC1–1990). These countries were severely affected by the spread of HIV/AIDs, deleterious socioeconomic and political conditions, conflicts, and wars during the 1990s [4, 5, 73]. This convergence club shows the worst longevity outcomes. 2. OECD countries (MC2–1990 and FC2–1990), mainly high-income, which are located in Europe, North and

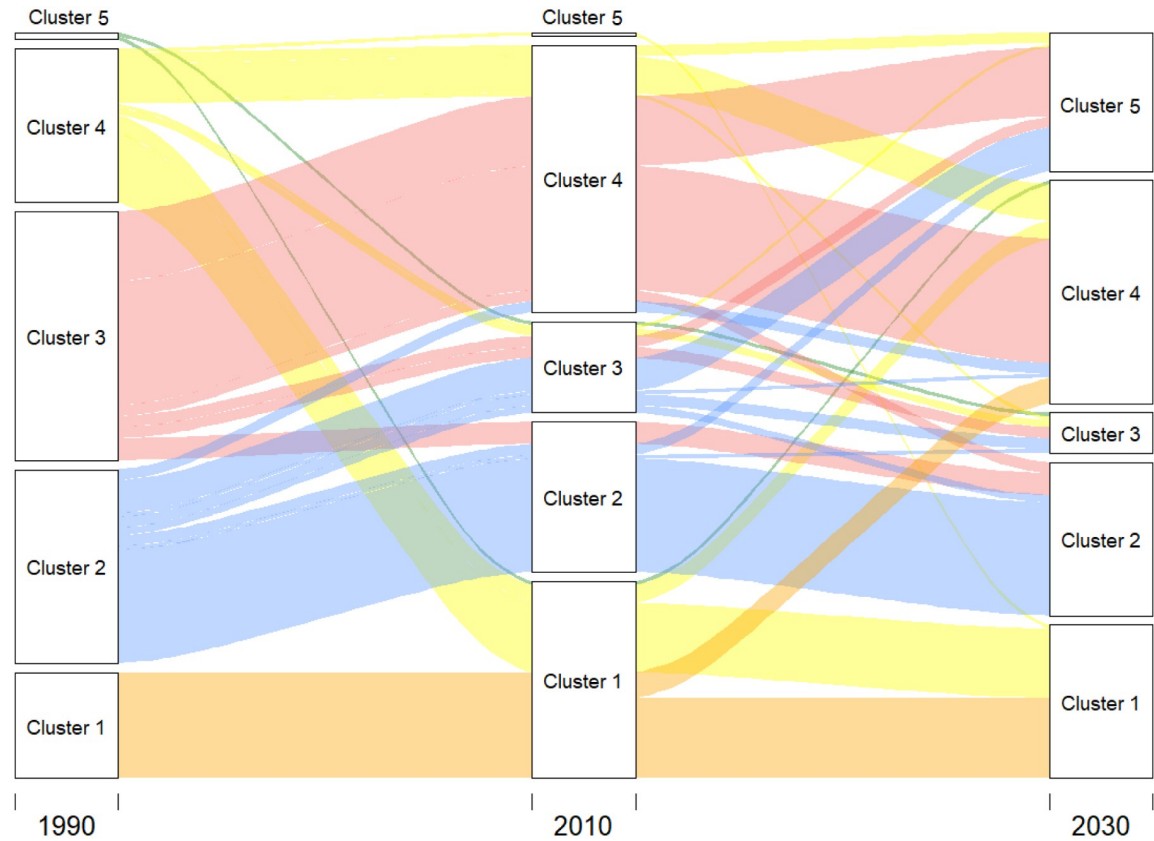

**Fig 8. Diagram to show how the convergence groups (clusters) change their female populations around the world over the periods of 1990–2010 and 2010–2030.** The image was created using *alluvial* R-package, developed by [70].

Central America, Japan, Chile, and the Australasia region. These countries have completed the epidemiological transition (see [5, 74]) also successful in the cardiovascular revolution [31] and therefore, they present the highest longevity outcomes. 3. China, North Africa, Asian Turkey, Latin America, and former URSS countries form another convergence club (MC3–1990 and FC3–1990), which includes middle-income countries and presents the second-best outcomes. Male populations from the former Soviet states suffered hectic mortality swings due to the fall of the Soviet Union [4, 74, 75]. This fact resulted in this region of Europe converging with middle-income rather than high-income countries. 4. South Africa, India, and the Pacific Islands, which are considered middle-income countries, converge in another club (MC4–1990 and FC4–1990). During 1950–1990 South Africa suffered gains in life expectancy [5], among other mortality indicators, producing a convergence in mortality with middle-income (developing) countries rather than converging in the Central African cluster with the worst mortality indicators in 1990. 5. Only two countries form Cluster 5. Rwanda is common for both sexes, the second one is Uganda for males and Qatar for females. It is noteworthy that this cluster contains only two countries for both genders. This observation results from the fact that the mortality indicators show a closer proximity to an independent centroid compared to the distance to any other four clusters. This fact is due to these countries' present irregularities in their mortality. Rwanda was affected by civil war, with a huge genocide of the Tutsi minority ethnic group that was killed by armed militias [76]. During 1990, Uganda was also affected by tensions with Sudan, Zaire, and Rwanda. There was a guerilla war which was led mainly by

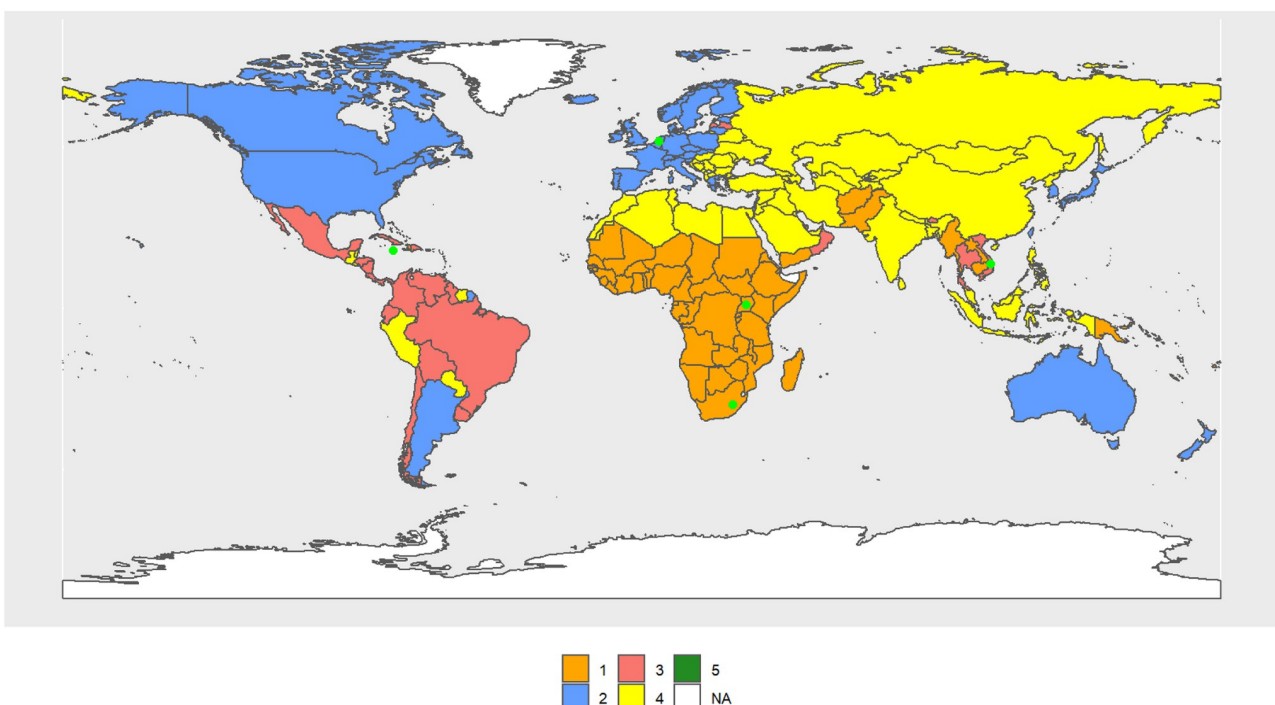

**Fig 9. World map of 2010 female clusters.** The map was created using *rnaturalearthdata* R-package, developed by [69].

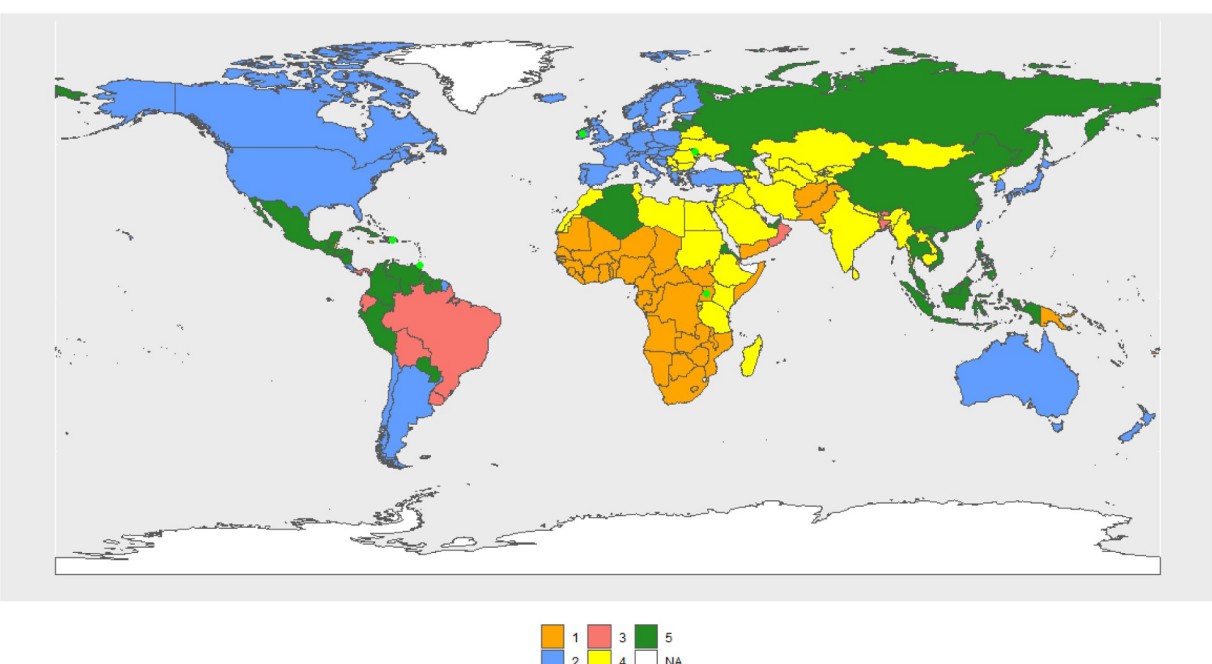

**Fig 10. World map of 2030 female clusters.** The map was created using *rnaturalearthdata* R-package, developed by [69].

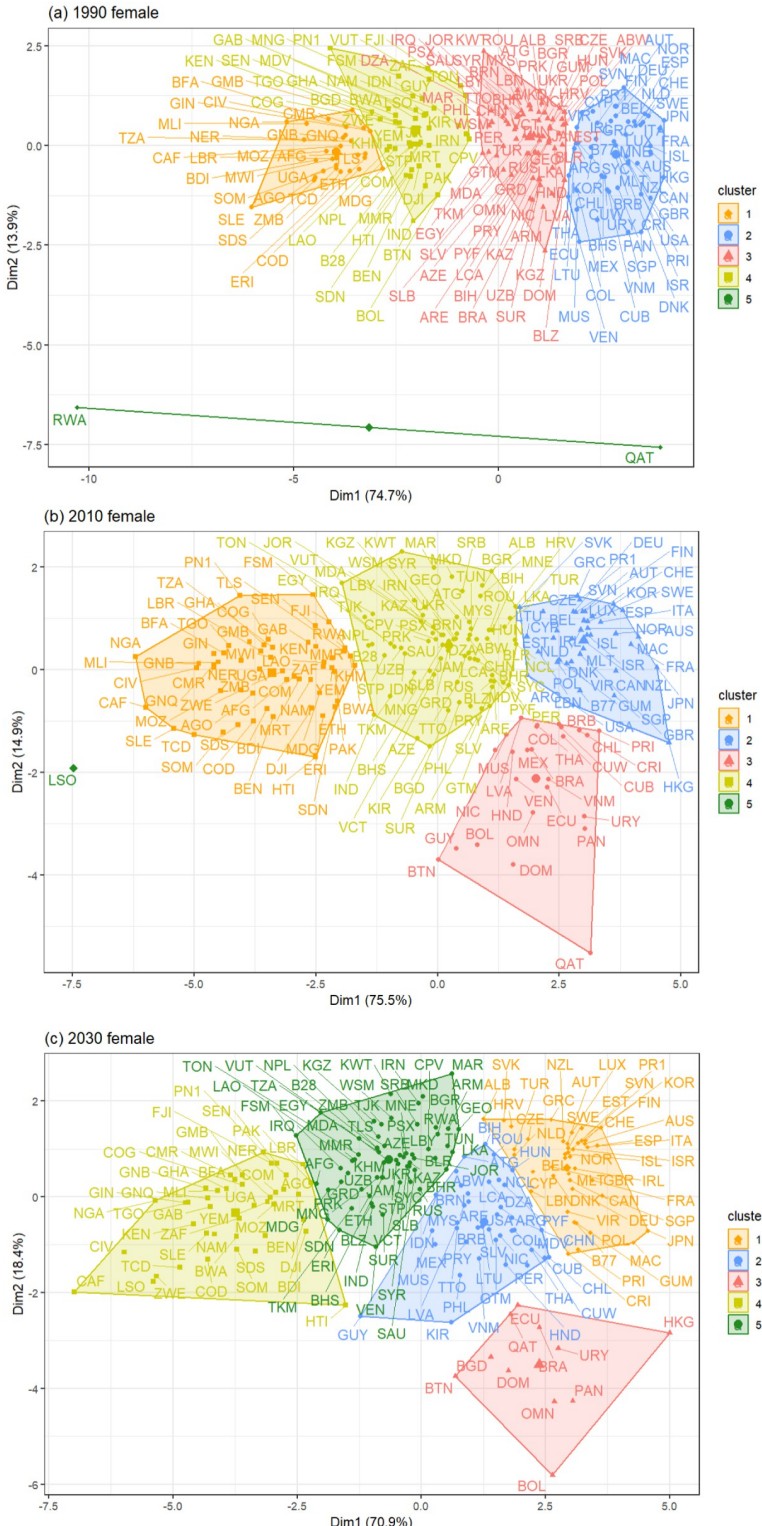

**Fig 11. Geographical proximity between the two principal components in (a) 1990, (b) 2010, and (c) 2030 for the mortality indicators in females using PCA.** The image was created using *factoextra* R-package, developed by [65].

Tutsi exiles in Uganda, known as the Rwandan Patriotic Front (RPF) [77]. Qatar was also involved in another war during the 90s and played an important role in the Gulf War [78].

Cluster 3 and 4, included in FC3–1990 for females and in MC4–1990 for males, are mainly different in Russia. Thus, Russian males converge to countries with the worst results, while females are presented in convergence clubs with better mortality outcomes. This fact is due to the worsening of male mortality during the early 1990s, see [4, 75]. This was determined by an increase in mortality from cardiovascular diseases and also from accidents and violent causes of death among adults from the 1970s to 1990s [34, 75].

In addition, within the discussion section, we aim to provide insights into the different stages of the epidemiological transition, described by [32], which are observed in each cluster during each time. To facilitate an understanding of each stage, we have included a brief summary of each stage in S4 Annex. In 1990, it is evident that MC1–1990 and FC1–1990 clusters are situated within the first stage of the epidemiological transition. This stage can be attributed to wars and widespread transmission of HIV and AIDs in Africa [74]. Likewise, the MC5–1990 and FC5–1990 clusters, consisting of only two countries, faced civil conflicts and also fell in the first stage. Contrasting, high-income countries in 1990, represented by MC2–1990 and FC2–1990, are in the third stage. This stage is characterized by a plateau in mortality, and the main cause of death is related to degenerative diseases (heart, cancer, and stroke) affecting individuals around the age of 70 [74]. Moreover, MC3–1990, FC3–1990, MC4–1990, and FC4–1990 clusters are situated in the second stage of the epidemiological transition. In this phase, deaths associated with epidemics and infectious diseases were abstinent. Instead, deaths are concentrated in middle age and are influenced by the Industrial Revolution and faced with chronic degenerative diseases.

It is interesting to note the main changes that are produced between the studied periods. As can be noted, the convergence clubs tend to resemble continents in 2010. Cluster 2 is formed by Australasia, Europe, and North America, which are high-income countries. African countries form Cluster 1, North Africa and Asia form Cluster 3, and Cluster 4 is made by Central and South America (Latin American countries). Furthermore, our results reveal that the main changes between 1990 and 2010 for both sexes are the following: all African countries, with the exception of the North, generate an independent convergence club. This can be partly explained by similar poverty levels, education, wars and spread of violence, gender inequalities, and fragility of the healthcare system, which facilitated the rapid spread of epidemics such as HIV/AIDS, malaria, and the Ebola virus [4, 79, 80], causing high premature mortality in young. South Africa, which had better mortality indicators in the 1990s, suffered a massive decline during the period 1990–2010 due to the exceptional nature of the HIV/AIDS epidemic [5]. Central and South America show a mortality process of convergence. Several authors have indicated that Latin American countries have experienced eroding effects caused by excess homicide mortality at younger adult ages, see [15, 81]. Asian and North African countries experience mortality convergence trajectories, as also noted by [5]. Male and female populations in former Soviet countries followed different changes in the convergence club. The male population converged to Asian and North African (middle-income) countries, while the female population converged to high-income countries with better mortality indicators. Male populations in former Soviet countries showed more problems than females, such as excessive alcohol consumption, lack of access to adequate health care, unemployment, poverty, and psycho-social stress [19, 82, 83]. Lesotho, in 2010, formed an independent cluster for males and females. This is due to Lesotho's most severe drought in 30 years, in 2008, and its higher HIV prevalence since the 1990s [84].

MC1–2010 and FC1–2010 remain in the first (initial) stage of the epidemiological transition. This fact is attributed to these countries' continued mortality stagnation, with the main

causes of deaths continuing to persist from the previous period (1990), including wars, epidemics, and infectious diseases [3]. High-income countries, as represented by MC2–2010 and FC2–2010, have progressed to the fourth stage, in comparison to the previous period. This stage is distinguished by a notable decline in mortality rates among advanced age groups, with degenerative diseases continuing to be the leading cause of death. MC3–2010, FC3–2010, MC4–2010, and FC4–2010 have advanced one stage in the epidemiological transition, arriving at the third stage. In this phase, mortality is stagnant, and the main causes of death differ between MC4–2010 and FC4–2010 (degenerative diseases) and MC3–2010 and FC3–2010 (external causes such as violent deaths and traffic accidents) [14].

Furthermore, we have shown how the convergence and divergence of mortality are expected to evolve in the future, in 2030. Understanding what the diverse mortality profiles will look like in the future can be important when considering the challenges of aging populations [85].

In 2030, the clusters are still configured as continents. While the main changes between convergence clubs during 2010 and 2030 are differences between the sexes. On the one side, the main changes for males will be as follows: the cluster formed by the African countries will be joined by several Asian countries with conflicts in the last years (2010–2020) such as Afghanistan, Kazakhstan, Kyrgyzstan, Pakistan, Turkmenistan and the Ukraine. These regions of Asia and Europe tend to converge with low-income countries where deleterious socioeconomic and political conditions, conflicts, and wars cause a large number of deaths, mainly in the African region. The convergence club of high-income countries will increase by seven in 2030 from different parts of the world: Algeria, Antigua and Barbuda, Chile, Puerto Rico, Estonia, Maldives, and Turkey. Projections suggest that these populations will continue to make further gains in mortality indicators to converge to the club with the best outcomes. For instance, Chile currently exhibits very high levels of development that are leading to better mortality indicators. Therefore, if this situation is maintained during the 21st century, it will lead to different regions of the world converging with the club with the best mortality outcomes. Latin America formed an independent convergence club with other small countries, which, in 2030, will reduce seven countries with respect to 2010, according to the results found by [81] that Latin American countries were converging towards the mortality regimes of the developed world in the first 15 years of 20th century. Hence, in 2030, these countries are projected to have and will obtain, the second-best mortality indicators after high-income regions. The convergence club in 2010, consisting of Asia, North Africa, and Asian Turkey, will be joined by several former Soviet countries (including Croatia, Bosnia-Herzegovina, Moldova, Montenegro, Slovakia, Latvia, Hungary, and Georgia) and three Latin American countries (Uruguay, Venezuela, and Argentina). Thus, middle-income countries will be expected to follow similar mortality trajectories. Former Soviet countries will converge to this club because projections consider that problems such as excessive alcohol consumption, lack of access to adequate health care, unemployment, poverty, and psycho-social stress problems will continue in the future. During the 2020s, several Latin American countries had a concentration of high levels of homicides, accidents, and suicides mainly in the male middle age population [81]. This fact could lead in the future to converge in the group of countries with middle-income and middle-longevity indicators (compared with the rest of the populations studied in this paper). Finally, the Central African Republic, Côte de I'voire, Lesotho, and Zimbabwe will converge to an independent club. These countries were in different wars and had deleterious socioeconomic and political conditions during 2010–2020. The projection took into account that all countries in this club reduced their indicator in the last two periods of the sample. Thus, the clustering method tends to create a convergence club with the worst mortality outcomes.

Regarding females, the African club cluster will follow a similar trend as males. The number of countries in 2010 has decreased by 23% compared with 2030. Furthermore, it is interesting to note that this convergence club, which is formed by the lowest mortality indicators, has a reduced number of countries compared with the male population. The European countries, Japan, and Australasia converged into an independent club with the best longevity indicators (OECD countries). The USA and several Former Soviet countries formed the same convergence club in 2010, while in 2030, they will form an independent one, as can be seen below. Mainly South American countries converge to an independent club formed by 11 countries. This projection considers that these countries will have the second-highest longevity indicators. It should be noted that the female population in South America presents a high longevity outcomes compared with males, see [86], according to the results found by [81] that Latin American countries are converging towards the mortality regimes of the developed world. This fact will make this region of the world follow the mortality steps of high-income countries (with the best mortality outcomes). Asia, North Africa, and East Africa are a convergence club mainly formed by middle-income countries. It is interesting to note that these projections consider that several countries in Africa, such as Eritrea, Ethiopia, Madagascar, Rwanda, Sudan, Tanzania, and Zambia, will present better results compared with the period of 2010, causing a movement toward this convergence club. This fact, which is mainly produced in the female population, is due to better socioeconomic and political conditions during the 2010s, leading to projections of better mortality indicators than in low-income countries. Latin American countries, the USA, China, Algeria, and several former Soviet countries converge into an independent cluster with the third-best longevity outcomes. The expansion of drug consumption is increasing the deaths among middle-aged individuals, which is observed in some sectors of the American male and female populations during the first and second decade of the 21st century [87]. This suggests that the USA will not converge with countries with the best mortality outcomes and therefore it will develop an independent convergence club with other countries of the world. In fact, this club has the third-best mortality outcomes, being very close to the second club.

The determination of the epidemiological transition stage for each group in the 2030 period requires careful consideration, as it involves projections into the future. The use of alternative mortality models can produce large differences in the forecasting process and, consequently, impact the clusters. Nevertheless, we seek to provide an idea of the stages expected for each cluster. Our cluster methodology assumes that the conflicts observed in 2010 continue into 2030. Hence, MC5–2030 is projected to remain within the initial stage of the epidemiological transition, affected by wars. MC1–2010 and FC1–2030 are expected to advance one stage further than the previous period, moving to the second stage. At this stage, mortality is characterized by a shift away from deaths related to wars, epidemics, and infectious, to focus more on middle-aged individuals dealing with chronic degenerative diseases. The third stage of the epidemiological transition will be included in MC3–2030 and FC3–2030 clusters. This stage assumes an equilibrium in mortality, where the main deaths are related to degenerative diseases, such as heart, cancer, and stroke in individuals around 70. Latin American countries, represented by MC3–2030 and FC3–2030 clusters, are projected to reach the fourth stage, joining with MC2–2030 and FC5–2030. This stage is characterized by concentrated in elderly age groups, while degenerative causes persist as the primary risk factors for mortality. Finally, FC2–2030 is considered to potentially enter the fifth stage where individuals are focused on exploring the bounds of their potential longevity.

The mortality data used have been taken from one database (WPP), in order to homogenize all available information. Nevertheless, and being the best existing mortality data from all over the world, some of the data may have weaknesses and compromise the quality of the

estimations. It should be noted that the mortality data used corresponds to the starting period of 1990. As noted during the paper, the predictions carried out for 2030 were based on the central estimation of the [38] model. So, we consider that the past mortality experience of each population (male and female) will continue in the future separately. However, it must be mentioned that year-to-year projections may change drastically due to unexpected and catastrophic events (pandemics, wars, natural phenomena, etc.). Therefore, the projections made in this study should be taken with caution.

It is interesting to highlight that three convergence clubs remain constant during the studied periods among males and females. In fact, these groups correspond to high-income clubs (Europe, Australasia, North America, and Japan), middle-income clubs (Asia and North Africa), and low-income clubs (Central and South Africa). In the high-income club, there are 29 countries in the male cluster and 34 in the female cluster, which remain constant during the studied periods; in the middle-income club, there are 12 countries in the male cluster and 10 in the female cluster; and in the low-income club, there are 22 countries in the male cluster and 22 in the female cluster. This fact is because, in high-income clubs, all countries have passed their epidemiological transition; in middle-income clubs, the transition is not over yet; and in low-income clubs, this transition is not just starting, although deleterious socioeconomic and political conditions are slowing down the process. The rest of the studied countries of the world will change their convergence clubs between periods and even develop their own convergence clubs, such as Latin American countries.

To analyze the global convergence among the countries during the two periods 1990 to 2010 and 2010 to 2030, we have decided to use $\beta$ regression [23, 88] as presented in Table 8 of S3 Annex (we also estimate the standard deviation among the indicators and countries and we have obtained consistent results). The outcomes show that, for all studied periods, indicators for both men and women exhibit the same behaviour (convergence or divergence) when $\beta$ is statistically significant. Specifically, we can state there is always convergence ($\beta < 0$) in $e_{0,t}$ and $GI_{0,t}$ with a level of significance of .01. Moreover, indicators such as M, $s_{0,t}$ and $y_{50\%;0,t}$ demonstrate convergence over time and across countries, with the exception of males during the period from 2010 to 2030. However, $e_{65,t}$ and $y_{70\%;65,t}$ diverge ($\beta > 0$) among all the periods and genders, with at least at a statistical level of significance of .10. In general, our analysis suggests that countries converge on indicators measured at birth (which include all ages) and diverge on indicators measured at 65 years of age (which include only older ages).

## 7 Conclusions

All convergence clubs and countries studied using the in-sample and out-of-sample approach have improved their mortality indicators. This fact clearly presents the growth in the ageing processes around the world during the last 30 years and even this process will continue in the future, according to our projections. Among all convergence clubs, Africa is the region with the most significant improvements in mortality indicators. Even the best-performing (high-income) countries continue to grow, although these improvements slowed over time. It should be noted that the Gini index works in the opposite way to the other mortality indicators. In other words, this indicator improves when the value decreases.

The male-female gap is decreasing among the convergence clubs and countries. These sex differences can be mainly determined by the harmful lifestyles of blue-collar males, see [89, 90]. In the future, this gap will narrow, although sex differences will continue to exist in the future. This fact can be explained by the Y chromosome, which is associated with an increased risk of mortality and age-related diseases, such as cardiovascular and heart diseases, fibrosis, cardiac dysfunction, etc. Indeed, as noted by [91], males lose this chromosome during their

ageing process while females keep it, allowing us to explain why these differences will continue to remain in the future, according to our projections.

In summary, this paper has updated the literature about world convergence clubs of mortality using a more sophisticated technique, such as the clustering method, and has also provided a global view of longevity. The large majority of papers about convergence/divergence in mortality have focused on how these clubs have been grouped in the past. Meanwhile, in this study, we presented that picture including a global view of mortality, throughout seven mortality indicators, and also included a perspective of how these convergence clubs could evolve in the future for male and female populations.

It should be noted that our study uses abridged life tables from the [39] database. This is one of the weaknesses of the manuscript, we consider mortality data from WPP and hence, the mortality indicators can exhibit reduced accuracy, especially in African countries. This fact is due to these data being produced with model life tables. However, we decided to use this database for an ambitious study of mortality convergence and divergence covering a wide range of countries around the whole world. Therefore, future research may consider repeating this analysis for developed countries that have complete mortality data by age to get a really accurate study.

Study limitations should be mentioned. It is difficult to anticipate the future behavior of morality using a model based on past trends, which do not always accurately reflect the evolution of future mortality behavior. As a future line of research, it would be particularly interesting to review our mortality estimates and cluster configurations in 2030, when we will have reliable data. This future analysis would allow us to assess the degree of accuracy of our 2023 estimates and the performance of the Lee-Carter model in projecting mortality trends and indicators.

Another future line of research, that we would like to explore, is to study the relationship between these nine indicators among the different life table models. Another possible avenue for future research is to consider DHS surveys in the mortality estimates.

## Supporting information

**S1 Annex.**
(PDF)

**S2 Annex. Estimation of mortality indicators.**
(PDF)

**S3 Annex. Supplementary information on mortality indicators.**
(PDF)

**S4 Annex. Brief summary of the stages of demographic transition.**
(PDF)

## Acknowledgments

The authors are indebted to the anonymous referees for their careful review of the manuscript and valuable remarks.

## Author Contributions

**Conceptualization:** David Atance, M. Mercè Claramunt, Xavier Varea, Jose Manuel Aburto.

**Data curation:** David Atance, M. Mercè Claramunt, Xavier Varea, Jose Manuel Aburto.

**Formal analysis:** David Atance, M. Mercè Claramunt, Xavier Varea, Jose Manuel Aburto.

**Funding acquisition:** David Atance, M. Mercè Claramunt, Xavier Varea, Jose Manuel Aburto.

**Investigation:** David Atance, M. Mercè Claramunt, Xavier Varea, Jose Manuel Aburto.

**Methodology:** David Atance, M. Mercè Claramunt, Xavier Varea, Jose Manuel Aburto.

**Project administration:** David Atance, M. Mercè Claramunt, Xavier Varea, Jose Manuel Aburto.

**Resources:** David Atance, M. Mercè Claramunt, Xavier Varea, Jose Manuel Aburto.

**Software:** David Atance, M. Mercè Claramunt, Xavier Varea, Jose Manuel Aburto.

**Supervision:** David Atance, M. Mercè Claramunt, Xavier Varea, Jose Manuel Aburto.

**Validation:** David Atance, M. Mercè Claramunt, Xavier Varea, Jose Manuel Aburto.

**Visualization:** David Atance, M. Mercè Claramunt, Xavier Varea, Jose Manuel Aburto.

**Writing – original draft:** David Atance, M. Mercè Claramunt, Xavier Varea, Jose Manuel Aburto.

**Writing – review & editing:** David Atance, M. Mercè Claramunt, Xavier Varea, Jose Manuel Aburto.

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
