## [Decision Letter · Decision Letter 0]

11 Sep 2023

PONE-D-23-11934Convergence and Divergence in Mortality: A Global Study From 1990 to 2030PLOS ONE

Dear Dr. David Atance,

Thank you for submitting your manuscript to PLOS ONE. After careful consideration, we feel that it has merit but does not fully meet PLOS ONE’s publication criteria as it currently stands. Therefore, we invite you to submit a revised version of the manuscript that addresses the points raised during the review process.

We look forward to receiving your revised manuscript.

Kind regards,

Jayanta Kumar Bora, PhD

Academic Editor

PLOS ONE

Journal Requirements:

"This work was partially supported by the funded chair UB-Longevity Institute."

"This work was partially supported by the funded chair UB-Longevity Institute."

"This work was partially supported by the funded chair UB-Longevity Institute."  

"No potential conflict of interest was reported by the author(s)."

5. One of the noted authors is a group or consortium "D.Atance, M. Claramunt, X. Varea, J.M. Aburto". In addition to naming the author group, please list the individual authors and affiliations within this group in the acknowledgments section of your manuscript. Please also indicate clearly a lead author for this group along with a contact email address.

6. We note that Figures 1, 2, 3, 5, 6 and 7 in your submission contain map images which may be copyrighted. All PLOS content is published under the Creative Commons Attribution License (CC BY 4.0), which means that the manuscript, images, and Supporting Information files will be freely available online, and any third party is permitted to access, download, copy, distribute, and use these materials in any way, even commercially, with proper attribution. For these reasons, we cannot publish previously copyrighted maps or satellite images created using proprietary data, such as Google software (Google Maps, Street View, and Earth). For more information, see our copyright guidelines: http://journals.plos.org/plosone/s/licenses-and-copyright.

                1. You may seek permission from the original copyright holder of Figures 1, 2, 3, 5, 6 and 7 to publish the content specifically under the CC BY 4.0 license. 

Reviewers' comments:

Reviewer's Responses to Questions

**Comments to the Author**

1. Is the manuscript technically sound, and do the data support the conclusions?

Reviewer #1: Yes

Reviewer #2: No

2. Has the statistical analysis been performed appropriately and rigorously? 

Reviewer #1: Yes

Reviewer #2: No

3. Have the authors made all data underlying the findings in their manuscript fully available?

Reviewer #1: Yes

Reviewer #2: No

4. Is the manuscript presented in an intelligible fashion and written in standard English?

Reviewer #1: Yes

Reviewer #2: No

5. Review Comments to the Author

Reviewer #1: This study analysed the convergence and divergences in mortality between 1990 and 2030 by analysing multiple mortality indicators at different ages. The authors used innovative clustering methods to measure global convergence and divergence in mortality. The authors provided a good background for this study and explained each indicator used to analyse past or future convergence and divergence. This paper will be a good contribution to the literature on convergence or divergence in health and mortality. This paper fits well for PLOS ONE readers. There are comments to improve its clarity for the readers.

1. The B Annex 2 explains the estimation of the mortality indicators; however, the explanation for some variables used in the equations needs to be included. The authors advised giving details of each component in the equation.

2. The authors mentioned that the cluster methods allow the splitting of the mortality indicator into different groups so that countries within each cluster are quite similar to each other. In Table 4, Rwanda and Qatar are the two countries that represent the FC5-1990 cluster. These two countries differed in mortality levels and exhibited large differences in females' life expectancy during 1990. The authors suggested talking about it in the results.

3. The authors provided the cluster characteristics in Tables 4, 5, and 6. The authors advised adding the number of countries to each cluster.

4. The discussion sections of the paper mainly discuss the results; the authors advised adding more about whether there is any similarity in epidemiological and demographic transitions across the countries within cluster groups.

Reviewer #2: This article aims to update the literature on "convergence or

divergence in mortality" with, in its abstract, references to Wilson

2001 and Mesle 2004, but it is not.

Both already published articles consider the evolution since

a much older and longer period, namely for developed countries the end of the

18th century and they are not proposing mortality projections.

Moreover, they essentially only use the level of mortality, namely

life expectancy, and are not interested at other functions which can be derived from a life table.

But, the previous authors are also looking at other sources of data which

is not the case for the proposed article. Meslé et al. will look for

data by major cause of death to explain the divergence, for example of

Eastern countries.

The proposed article looks at only one source, not even discussing the

quality of the data neither on how they are constructed.

The article is a good statistical exercise, using the latest published

methods in demographic and statistical R packages but on a data set

which has been built by the UN for a less ambitious aim.

It seems to us that the authors are seeking to compensate the lack of

knowledge on demographic data sources by adding all the "modern"

indicators derived from a unique abridged life table.

Let us yet describe their work which could be of interest if it was

limited to countries where mortality rates are not already a result of

a model.

...

(see long attachment)

6. PLOS authors have the option to publish the peer review history of their article (what does this mean?). If published, this will include your full peer review and any attached files.

Reviewer #1: No

Reviewer #2: **Yes: **Nicolas Brouard

---

## [Author Response · Author response to Decision Letter 0]

14 Oct 2023

Dear Reviewers from Financial Innovations,

We enclose the revised version of the manuscript entitled “Convergence and Divergence in Mortality: A Global Study From 1990 to 2030” (ref. PONE-D-23-11934) in which we have incorporated the referees' changes and suggestions in red color.

We would like to thank the reviewers and the editor for their insightful comments that have allowed us to improve the original version of the paper. All the comments have

been carefully followed and changed texts corresponding to major changes are red colored in the revised manuscript. Please see the attached files our point-by-point responses from each reviewer. In fact, these files are attached at the end of the pdf as "Reviewer1_PlosOne.pdf" and "Reviewer2_PlosOne.pdf".

Yours sincerely,

David Atance, Mercè Claramunt, Xavier Varea and José Manuel Aburto.

---

## [Decision Letter · Decision Letter 1]

1 Dec 2023

Convergence and Divergence in Mortality: A Global Study From 1990 to 2030

PONE-D-23-11934R1

Dear Dr. David Atance,

We’re pleased to inform you that your manuscript has been judged scientifically suitable for publication and will be formally accepted for publication once it meets all outstanding technical requirements and subject to incorporate the reviewer's new comments.

Kind regards,

Jayanta Kumar Bora,PhD

Academic Editor

PLOS ONE

Additional Editor Comments (optional):

Reviewers' comments:

Reviewer's Responses to Questions

**Comments to the Author**

1. If the authors have adequately addressed your comments raised in a previous round of review and you feel that this manuscript is now acceptable for publication, you may indicate that here to bypass the “Comments to the Author” section, enter your conflict of interest statement in the “Confidential to Editor” section, and submit your "Accept" recommendation.

Reviewer #2: (No Response)

2. Is the manuscript technically sound, and do the data support the conclusions?

Reviewer #2: Yes

3. Has the statistical analysis been performed appropriately and rigorously? 

Reviewer #2: Yes

4. Have the authors made all data underlying the findings in their manuscript fully available?

Reviewer #2: Yes

5. Is the manuscript presented in an intelligible fashion and written in standard English?

Reviewer #2: Yes

6. Review Comments to the Author

Reviewer #2: As requested in my firt referee, the authors did an excellent job in adding a principal component analysis and mixing its results with the cluster analysis.

And as expected, the different clusters are completely distinct; this beautiful representation is the most interesting part of the article for a demographer. The authors could have graduated the first component in terms of life expectancy (e0) on their graphs, in order to allow readers to appreciate not only the different levels 40, 50, 60, 70, 80 years for a specific period (like 1990) but also to judge improvements in terms of life expectancy in 2010 and 2030. A graduation of the life expectancy at 65 for developed countries could also be of interest to a reader. Of course, these scales are not linear.

A plot of the second component and the third component (i.e. independent of the overall mortality level), would help readers understand the correlation matrix analysis.

You could limit yourself to representing the 9 variables on the correlation circle and overlay only a few typical countries from each cluster. These new graphs, tables and this paragraph on correlations are particularly useful because they allow non-demographers to understand the mortality transition: when infant and child mortality are high, e(65) and most other variables are useless; but, after the transition, they begin to play a role. The authors added the necessary remarks on the quality of the UN sources, even if they did not mention the absence of vital statistics (the only source that can provide independent data by age) in too many countries.

I also appreciate the efforts to make tables and graphs more readable.

7. PLOS authors have the option to publish the peer review history of their article (what does this mean?). If published, this will include your full peer review and any attached files.

Reviewer #2: **Yes: **Nicolas Brouard, French Institute for Population Studies (INED, Paris)

---

## [Editor Report · Acceptance letter]

5 Dec 2023

PONE-D-23-11934R1 

Convergence and Divergence in Mortality: A Global Study
From 1990 to 2030 

Dear Dr. Atance:

I'm pleased to inform you that your manuscript has been deemed suitable for publication in PLOS ONE. Congratulations! Your manuscript is now with our production department. 

Kind regards, 

on behalf of

Dr. Jayanta Kumar Bora 

Academic Editor

PLOS ONE